# Hypoxic Preconditioned Nanofat at 1% O_2_ for 24 h Loses Its Regenerative In Vivo Vascularization Capacity

**DOI:** 10.3390/cells15020100

**Published:** 2026-01-06

**Authors:** Francesca Bonomi, Ettore Limido, Andrea Weinzierl, Caroline Bickelmann, Emmanuel Ampofo, Yves Harder, Matthias W. Laschke

**Affiliations:** 1Institute for Clinical and Experimental Surgery, Saarland University, PharmaScienceHub (PSH), 66421 Homburg, Germany; francescabonomi.bonnie@gmail.com (F.B.); limidoettore@gmail.com (E.L.); andreaweinzierl@icloud.com (A.W.); caroline.bickelmann@uks.eu (C.B.); emmanuel.ampofo@uks.eu (E.A.); 2Department of Plastic Surgery and Hand Surgery, University Hospital Zurich, 8006 Zurich, Switzerland; 3Department of Plastic, Reconstructive and Aesthetic Surgery and Hand Surgery, Centre Hospitalier Universitaire Vaudois (CHUV), 1005 Lausanne, Switzerland; yves.harder@chuv.ch; 4Faculty of Biology and Medicine, University of Lausanne (UNIL), 1005 Lausanne, Switzerland

**Keywords:** hypoxic preconditioning, nanofat, dermal substitutes, regeneration, survival, angiogenesis, vascularization, inflammation

## Abstract

**Highlights:**

**What are the main findings?**
Hypoxic preconditioning at 1% O_2_ for 24 h activates nanofat and shifts its protein expression profile towards a pro-angiogenic phenotype without affecting its viability.The applied hypoxic preconditioning protocol does not improve the in vivo vascularization capacity of nanofat after its seeding on implanted dermal substitutes.

**What are the implications of the main findings?**
Hypoxic preconditioning at 1% O_2_ for 24 h may be too stressful for nanofat that is subsequently exposed to prolonged in vivo hypoxia.Milder preconditioning protocols should be alternatively tested in future studies.

**Abstract:**

Hypoxic preconditioning is increasingly explored to enhance the survival and vascularization of fat grafts. In this study, nanofat from donor mice was exposed to hypoxia (1% O_2_) for 24 h to investigate the effects of this preconditioning protocol on the viability, gene expression and vascularization capacity of this mechanically processed fat derivative. Ex vivo analyses revealed that hypoxic preconditioning does neither affect apoptotic nor necrotic cell death within nanofat but significantly upregulates the expression of hypoxia-inducible factor (HIF)-1α and stromal cell-derived factor (SDF)-1 compared to non-preconditioned nanofat. Moreover, preconditioned nanofat exhibited a pro-angiogenic protein expression profile. For in vivo analyses, dermal substitutes were either seeded with preconditioned or non-preconditioned nanofat and transferred into dorsal skinfold chambers of mice to assess their vascularization by intravital fluorescence microscopy. Unexpectedly, implants seeded with preconditioned nanofat exhibited a significantly reduced functional microvessel density when compared to non-preconditioned controls. Immunohistochemical analyses also confirmed a lower microvessel density within the implants of the preconditioned group. These findings suggest that hypoxic preconditioning at 1% O_2_ for 24 h cannot be recommended for enhancing the regenerative in vivo vascularization capacity of nanofat. Therefore, milder preconditioning protocols with shorter periods of hypoxia or higher oxygen levels should be alternatively tested in future studies.

## 1. Introduction

Preconditioning is a strategy, which involves the exposure of cells or tissues to different stimuli to induce adaptive mechanisms that enhance their resistance to subsequent stressors. This process leads to the activation of survival pathways, the upregulation of stress-response genes and the secretion of cytokines, ultimately improving cellular viability [1,2]. Several preconditioning methods, including hypoxic, pharmacological and mechanical approaches, have been tested for various clinical applications [3]. Among these, hypoxic preconditioning, i.e., transient exposure to reduced oxygen levels, has shown beneficial effects in cardiovascular surgery, organ transplantation and plastic surgery, where it is also known as the ‘delay phenomenon’ [4,5,6,7]. Hypoxic preconditioning is known to stabilize hypoxia-inducible factor (HIF)-1α, which stimulates blood vessel formation by the expression of vascular endothelial growth factor (VEGF) and enhances cell survival by shifting metabolism towards glycolysis [8,9,10,11,12].

Recent studies investigated hypoxic preconditioning of autologous adipose tissue to improve the survival rates of fat grafts. Evidence from animal experiments suggests that short-term hypoxic exposure before transplantation by means of tourniquet application (also known as remote preconditioning) or adipose flap elevation before fat harvesting enhances the long-term survival of adipose tissue grafts [13,14]. Moreover, hypoxia-mimetic agents, such as deferoxamine, have been used to replicate these effects pharmacologically, further supporting the potential of this strategy in optimizing fat graft survival [15].

Fat grafting is a widely used technique in reconstructive and esthetic surgery, which involves the harvesting and reinjection of autologous adipose tissue to restore volume and/or promote tissue regeneration. This procedure has been used effectively for various indications from scar remodeling and soft tissue augmentation to the management of chronic wounds and fistulas [16,17,18,19]. In recent years, different processing techniques have been developed to further enhance the regenerative properties of adipose tissue. Notably, nanofat is a mechanically emulsified and filtered adipose derivative that contains a low number of mature adipocytes while preserving adipose-derived stem cells (ASCs), microvascular fragments and growth factors [20,21,22]. Due to these components with a strong regenerative potential, nanofat was effective in the treatment of chronic wounds, skin aging, scars and androgenic alopecia [23,24,25,26]. However, despite the increasing use of nanofat in clinical practice, the effects of hypoxic preconditioning on this mechanically processed fat derivative have not yet been analyzed.

Accordingly, the objective of this study was to investigate the effects of hypoxic preconditioning on the viability of nanofat and its expression of angiogenesis-related factors. Furthermore, the regenerative in vivo vascularization capacity of nanofat was tested in a well-established mouse model [27].

## 2. Materials and Methods

### 2.1. Animals

The in vivo experiments were conducted following the National Institutes of Health (NIH) Guidelines on the Care and Use of Laboratory Animals (NIH publication #85-23 Rev. 1985) and the European legislation on the protection of animals (Directive 2010/63/EU). Protocols were approved by the local authorities (permission number: 06-2022; State Office for Consumer Protection, Saarbrücken, Germany).

For ex vivo analyses, subcutaneous inguinal adipose tissue was harvested from C57BL/6J wild-type mice (Institute for Clinical and Experimental Surgery, Saarland University, Homburg, Germany) with an average age of 4 months and a body weight of 25 g. For in vivo experiments, C57BL/6-Tg (CAG-EGFP)131Osb/LeySopJ mice (The Jackson Laboratory, Bar Harbor, ME, USA) with an average age of 4 months and a body weight of 30 g served as donors of green fluorescent protein (GFP)^+^ adipose tissue. C57BL/6J wild-type mice with an average age of 4 months and a body weight of 25 g were equipped with dorsal skinfold chambers and housed individually to prevent mutual injuries due to the chambers at a temperature of 22–24 °C, a relative humidity of 50–60% and a 12 h light/dark cycle for the entire duration of the experiment. All mice had free access to tap water and pellet chow (Altromin, Lage, Germany).

### 2.2. Anesthesia

At the beginning of all surgical procedures, the mice received an intraperitoneal injection (i.p.) of ketamine hydrochloride (100 mg/kg; Ketabel^®^; Bela-pharm GmbH & Co. KG, Vechta, Germany) and xylazine (12 mg/kg; Rompun^®^; Bayer, Leverkusen, Germany). Perioperative pain management was performed by subcutaneous administration of carprofen (10 mg/kg; Rimadyl^®^; Zoetis Deutschland GmbH, Berlin, Germany).

### 2.3. Generation and Hypoxic Preconditioning of Nanofat

Nanofat was generated as previously performed in [22]. For this, anesthetized donor mice were euthanized via cervical dislocation. Their inguinal subcutaneous fat pads were harvested, washed, minced and mechanically emulsified and filtered to obtain nanofat [22].

The nanofat from each donor mouse was split into two equal portions. The first aliquot (preconditioned nanofat; hypoxia) was exposed at 37 °C in a tri-gas incubator (CB-S 170, Binder, Tuttlingen, Germany) to hypoxic conditions (1% O_2_, 5% CO_2_) for 24 h. Thereafter, the preconditioned samples were analyzed ex vivo or used for in vivo experiments. The second aliquot (non-preconditioned nanofat; control) was not exposed to hypoxia but investigated ex vivo or in vivo directly after its generation.

### 2.4. Ex Vivo Analysis of Nanofat

The viability of control (*n* = 4) and preconditioned nanofat (*n* = 4) was analyzed by flow cytometric measurements [28]. Briefly, individual cells obtained by enzymatic dissociation of nanofat with Accutase^®^ (BioLegend, Fell, Germany) were washed, resuspended in incubation buffer and stained for 15 min with propidium iodide (50 μg/mL; BD Biosciences, Heidelberg, Germany) and annexin V (100 μg/mL; ImmunoTools, Friesoythe, Germany) according to the manufacturer’s protocol. Subsequently, the stained cells were analyzed using a FACSLyric (BD Biosciences) to quantify the fraction of vital, apoptotic, necroptotic and necrotic cells, which were expressed as the percentage of totally analyzed cells.

Moreover, total RNA was extracted from nanofat samples (control: *n* = 4; preconditioned nanofat: *n* = 4) using QIAzol lysis reagent (Qiagen, Hilden, Germany). The corresponding cDNA was generated using c-DNA Synthesis Kit (iScript cDNA Synthesis Kit; BioRad, Hercules, CA, USA) and subjected to quantitative real-time polymerase chain reaction (qRT-PCR) using SYBR-Green Supermix (SsoAdvanced Universal SYBR Green Supermix; BioRad) and a CFX96 RT-PCR System (BioRad). Murine β-actin served as control. The concentration of forward and reverse primers (dissolved in RNase/DNase-free H_2_O) was 500 nM. Primer sequences were: 5′-CGGCGACATGGTTTACATTT-3′ (forward) and 5′-TTTCTCACTGGGCCATTTCT-3′ (reverse) for HIF-1α; 5′-CCAACGTCAAGCATCTGAAA-3′ (forward) and 5′-AATTTCGGGTCAATGCACAC-3′ (reverse) for stromal cell-derived factor (SDF)-1.

For histological and immunohistochemical investigations, nanofat samples (control: *n* = 4; preconditioned nanofat: *n* = 4) were fixed in 4% formalin, embedded in paraffin and cut into 3 µm thick sections. Sections were stained with hematoxylin–eosin (HE) as well as with rabbit anti-cleaved caspase (Casp)-3 (1:100; Cell Signaling, Leiden, The Netherland) and rabbit anti-HIF-1α (1:100; Abcam, Cambridge, UK) antibodies. A biotinylated goat-anti-rabbit IgG antibody (ready-to-use; Abcam) served as secondary antibody. Subsequently, a BX53 microscope and the imaging software CellSens Dimension (version 1.11; Olympus, Hamburg, Germany) were used to quantitatively analyze the numbers of Casp-3^+^ and HIF-1α^+^ cells expressed as the percentage of totally analyzed cells.

Finally, a mouse angiogenesis proteome profiler array kit (R&D Systems, Bio-Techne; Wiesbaden-Nordenstadt, Germany) was used to compare angiogenesis-related protein expression profiles in the different nanofat samples (control: *n* = 4; preconditioned nanofat: *n* = 4), as previously described [28].

### 2.5. Seeding of Dermal Substitutes with Nanofat

Dermal substitutes (Integra^®^ single layer; 1.3 mm-thick; Integra Life Sciences, Gent, Belgium) were punched into discs with a diameter of 4 mm (biopsy punch; Kai Europe GmbH, Solingen, Germany) and incubated for 10 min in either control or preconditioned nanofat for proper seeding, as previously established [27].

### 2.6. Dorsal Skinfold Chamber Model

The impact of preconditioned and control nanofat on the vascularization and tissue integration of dermal substitutes was analyzed in the dorsal skinfold chamber model [27]. For this purpose, two titanium frames (Irola Industriekomponenten GmbH & Co. KG, Schonach, Germany) were surgically implanted onto the back of anesthetized C57BL/6J wild-type mice, following shaving and chemical depilation (asid-med depilation cream; Asid Bonz GmbH, Herrenberg, Germany). A circular area of skin including the panniculus carnosus muscle (~15 mm in diameter) was excised to expose the panniculus carnosus muscle and skin of the other side of the skinfold for later analyses through the observation window in the middle of the titanium frames. After 48 h of recovery, a dermal substitute seeded with control (*n* = 8) or preconditioned nanofat (*n* = 8) was implanted into the chamber, which was then sealed with a cover slip and snap ring.

### 2.7. Intravital Fluorescence Microscopy

The implants were repeatedly analyzed using intravital fluorescence microscopy over 14 days. For this purpose, the anesthetized animals received an intravenous retrobulbar injection of 0.05 mL fluorescein isothiocyanate (FITC)-labeled dextran (5%, 150,000 Da; Sigma-Aldrich, Taufkirchen, Germany) and 0.05 mL rhodamine 6 G (0.1%; Sigma-Aldrich) for the staining of blood plasma and leukocytes, respectively. Thereafter, the chambers were examined under a fluorescence epi-illumination microscope (Zeiss Axiotech; Carl Zeiss Microscopy, Oberkochen, Germany). The microscopic images were recorded with a charge-coupled device camera (Axiocam 702 mono; Carl Zeiss Microscopy), stored on an external hard drive and analyzed offline using CapImage (version 8.10.1; Zeintl, Heidelberg, Germany).

The quantitative analysis of the microscopic images included the assessment of the total number of perfused regions of interest (ROIs) (%), the functional microvessel density (cm/cm^2^) as well as microhemodynamic parameters (diameter (μm), centerline red blood cell (RBC) velocity (μm/s), shear rate (s^−1^) and volumetric blood flow (pL/s)) of individual microvessels in 8 ROIs (center: *n* = 4; border zones: *n* = 4) of each implant. Additionally, microhemodynamic parameters and leukocyte–endothelial cell interactions (rolling leukocytes (min^−1^) and adherent leukocytes (mm^−2^)) were measured in postcapillary and collecting venules within the host tissue next to the implants in 4 different ROIs [27].

### 2.8. Histological and Immunohistochemical Analysis of Implants

At the end of the in vivo experiments, the implants with the surrounding chamber host tissue were carefully excised, fixed in 4% formalin, embedded in paraffin and sectioned. HE stainings and immunostainings were performed, as previously described [28]. Briefly, antibodies against CD31, lymphatic vessel endothelial hyaluronan receptor (LYVE)-1 and GFP as well as collagen (Col) I, Col III, CD68, myeloperoxidase (MPO) and CD3 were used.

The microvessel density (mm^−2^), lymph vessel density (mm^−2^) as well as CD31^+^/GFP^+^ microvessels (%) and LYVE-1^+^/GFP^+^ lymph vessels (%) were quantitatively assessed in one representative section per sample using a BX53 microscope and the software cellSens Dimension (version 1.11; Olympus, Hamburg, Germany). Moreover, the numbers of CD68^+^ macrophages (mm^−2^), MPO^+^ neutrophilic granulocytes (mm^−2^) and CD3^+^ lymphocytes (mm^−2^) as well as the total Col I and Col III ratio (implant/skin) were analyzed in 4 ROIs (center: *n* = 2; border zones: *n* = 2) of each implant.

### 2.9. Statistical Analysis

Statistical analyses were performed using GraphPad Prism 10.1.2 (GraphPad Software, San Diego, CA, USA). The group sizes were chosen according to previous studies using the herein described model [27,28]. Following the 3R principle in animal testing, the number of animals per group was reduced to a minimum while guaranteeing sufficient statistical power (0.8) to detect biological meaningful differences. Data were first evaluated for normal distribution and equal variance. Comparison between two groups was conducted using the unpaired Student’s t-test for parametric data or Mann–Whitney rank sum test for non-parametric data. Corrections for multiple comparisons were not performed. Results were given as mean ± standard error of the mean (SEM). A *p*-value < 0.05 was considered statistically significant.

## 3. Results

### 3.1. Ex Vivo Analysis of Nanofat

Macroscopically, the consistency and coloration of preconditioned and control nanofat did not differ. Flow cytometric viability analyses demonstrated that hypoxic preconditioning at 1% O_2_ for 24 h does not affect apoptotic or necrotic cell death within nanofat when compared to non-preconditioned control (Figure 1A). Accordingly, immunohistochemical evaluation of nanofat samples showed a low rate of Casp-3^+^ apoptotic cells (< 1%) in both groups without statistical difference (Figure 1B,C).

RT-PCR revealed a marked upregulation of the mRNA expression of hypoxia-responsive HIF-1α and SDF-1 in preconditioned nanofat, indicating a strong transcriptional response to hypoxic stress (Figure 1D). In line with these results, preconditioned nanofat also contained more HIF-1α^+^ cells compared to control (Figure 1E,F).

To characterize the expression profile of angiogenesis-related factors within control and preconditioned nanofat, an in vitro proteome profiler angiogenesis array was conducted. This array revealed a clear trend towards a pro-angiogenic protein expression profile in preconditioned nanofat with 28 out of 39 pro-angiogenic factors being upregulated and 11 out of 14 anti-angiogenic factors being downregulated when compared to control (Table 1).

### 3.2. In Vivo Microscopy of Nanofat-Seeded Dermal Substitutes

To analyze the in vivo vascularization capacity of control and preconditioned nanofat, dermal substitutes were seeded with both nanofat types and implanted into dorsal skinfold chambers. This enabled the repeated assessment of implant vascularization by means of intravital fluorescence microscopy (Figure 2A,B). This process was characterized by the stepwise ingrowth of microvessels from the surrounding host tissue into the border zones of the implants. Quantitative analyses of the newly developing microvascular networks revealed a comparable fraction of perfused ROIs in the border zones of dermal substitutes seeded with control and preconditioned nanofat (Figure 2C). However, implants seeded with preconditioned nanofat exhibited a significantly lower functional microvessel density in their border zones between day 10 and 14 when compared to controls (Figure 2E). Almost no vascular ingrowth was observed in the center of both implant types throughout the 14-day observation period (Figure 2D,F). Accordingly, microhemodynamic parameters of individual microvessels, including diameter, centerline RBC velocity, shear rate and volumetric blood flow, were only measured in the border zones of the implants (Table 2). In contrast to dermal substitutes seeded with preconditioned nanofat, these parameters could already be measured on day 6 for dermal substitutes seeded with control nanofat due to their accelerated vascularization (Table 2). There were only minor differences between the two groups with a tendency towards improved blood perfusion of microvessels in the control group.

Postcapillary and collecting venules in the host tissue next to the implants provided comparable microhemodynamic conditions for the investigation of leukocytes (Table 3). This investigation did not reveal marked differences in the number of rolling or adherent leukocytes between dermal substitutes seeded with control or preconditioned nanofat (Figure 3A–C), indicating a comparable inflammatory response to the implants of the two groups.

### 3.3. Histological and Immunohistochemical Analysis of Nanofat-Seeded Dermal Substitutes

On day 14, the implanted nanofat-seeded dermal substitutes were processed for additional histological and immunohistochemical analyses. HE-stained sections revealed a comparable integration of the implants seeded with control and preconditioned nanofat into the surrounding tissue (Figure 4A,B). This was characterized by the formation of a dense granulation tissue in the border zones of the implants, which also reached into the outer pores of the dermal substitutes. The Col I and III content of this granulation tissue was comparable in both groups (Figure 4C–F). In contrast, the center of the implants mainly contained individual cells and only low amounts of extracellular matrix (Figure 4A–F).

In line with our in vivo microscopic results, dermal substitutes seeded with control and preconditioned nanofat markedly differed in terms of their vascularization. In fact, implants seeded with preconditioned nanofat presented with a significantly lower density of CD31^+^ microvessels in their border and center zones (Figure 5A–C). Moreover, GFP/CD31 double stainings demonstrated that almost no microvessels were GFP^+^ in these implants (Figure 5B,D). In contrast, implants seeded with control nanofat contained ~80% GFP^+^ microvessels, indicating their origin from the seeded nanofat of GFP^+^ donor mice (Figure 5A,D). Furthermore, immunohistochemical staining of LYVE-1 expression revealed a few lymph vessels in the border and center zones of dermal substitutes seeded with control nanofat (Figure 6A,B). Most of these lymph vessels were GFP^+^ (Figure 6C,D). In contrast, no lymph vessels were detectable in implants seeded with preconditioned nanofat (Figure 6A–D).

The quantification of immune cell infiltration via immunohistochemical CD68 (macrophages), MPO (neutrophilic granulocytes) and CD3 (lymphocytes) stainings revealed no significant differences between the two groups, indicating a comparable host immune response to implants seeded with control or preconditioned nanofat (Figure 7A–F).

## 4. Discussion

Hypoxic preconditioning has emerged as a promising strategy to enhance organ and tissue survival in cardiovascular surgery, organ transplantation, flap surgery and fat grafting [4,5,6,7]. This effect has mainly been attributed to the upregulation of HIF-1α under hypoxic conditions, which plays a crucial role in metabolic adaptation and induction of pro-angiogenic genes [8]. However, in this study, we found that hypoxic preconditioning of nanofat does not improve its regenerative vascularization capacity in vivo. In fact, we even found that dermal substitutes seeded with preconditioned nanofat exhibit a markedly reduced microvessel density when compared to controls.

Commonly tested hypoxic preconditioning protocols involve either intermittent or continuous exposure to low oxygen levels, with specific parameters varying depending on the target tissue and intended application. The most frequently applied protocols involve oxygen concentrations between 1% and 5% for various durations ranging from 10 min to 48 h [29,30,31,32,33]. Notably, a 24 h exposure to 1% oxygen has often been used for the preconditioning of stem cells [30,34,35,36]. Accordingly, we also used this approach for the preconditioning of nanofat, which is known to be a rich source of ASCs. The current ex vivo analyses proved that this preconditioning protocol does not affect the viability of nanofat. On the other hand, it effectively activated the tissue, as indicated by the upregulation of HIF-1α and SDF-1 expression. Moreover, hypoxic preconditioning shifted the protein expression profile of nanofat towards a pro-angiogenic phenotype.

Based on the promising ex vivo results, we assumed that hypoxic preconditioned nanofat may also improve the in vivo vascularization and tissue integration of seeded dermal substitutes. However, this was not the case. In fact, the in vivo experiments in the dorsal skinfold chamber model showed that dermal substitutes seeded with preconditioned nanofat exhibited a markedly reduced vascularization throughout the 14-day observation period compared to controls. This unexpected outcome may be explained by the fact that directly after implantation into the dorsal skinfold chamber, the implants lacked their own blood supply and were dependent on oxygen and nutrient diffusion from the surrounding vessels. Therefore, we hypothesize that in addition to the ex vivo hypoxic preconditioning period of 24 h, the cells within the seeded nanofat may have still suffered from prolonged hypoxia under in vivo conditions, resulting in the loss of their intrinsic vascularization capacity. In line with this view, Jian et al. [37] reported that exposure of endothelial progenitor cells to hypoxia for 24 h promotes their motility and tube formation, while these beneficial effects are reversed by prolonged hypoxia for 48 and 72 h. Moreover, it is well known that long hypoxic phases are typically associated with mitochondrial dysfunction, increased reactive oxygen species (ROS) formation and the accumulation of deleterious metabolites, resulting in cellular injury and final cell death [38]. An excessive activity of the HIF pathway may have additionally led to maladaptive outcomes, such as the induction of apoptosis [39]. In fact, the effects of HIF-1 signaling have been described as a double-edged sword, which may be particularly dependent on the degree and duration of hypoxia [39].

In previous studies we have shown that the intrinsic vascularization capacity of nanofat is markedly driven by microvascular fragments [22,27]. These microvascular fragments rapidly interconnect with each other and surrounding blood vessels to develop new blood-perfused microvascular networks within the implants [27,40,41]. Accordingly, we found that dermal substitutes seeded with control nanofat contained ~80% GFP^+^ microvessels, indicating their origin from the seeded nanofat. On the other hand, implants seeded with preconditioned nanofat exhibited almost no GFP^+^ microvessels on day 14 after implantation into the dorsal skinfold chamber. Moreover, we have recently demonstrated that nanofat also contains lymphatic vessel fragments that actively contribute to lymph vessel formation [25]. Accordingly, we detected some LYVE- 1^+^/GFP^+^ lymph vessels in control nanofat-seeded dermal substitutes, whereas the hypoxic preconditioned group did not contain any GFP^+^ lymph vessels. These observations further support our assumption that prolonged hypoxia may have destroyed the blood and lymphatic vessels inside the preconditioned nanofat through the mechanisms mentioned before.

Finally, it should be considered that hypoxic preconditioning may have also negatively affected extracellular matrix proteins or altered the inflammatory activity of nanofat. In fact, HIF-1α has been shown to induce excessive deposition and abnormal cross-linking of collagen in adipose tissue, driving local fibrosis [42]. Moreover, hypoxia has been described as a cause of increased inflammation of adipose tissue [43]. Considering the important and complex link between fibrosis, inflammation and angiogenesis [44], we additionally evaluated whether hypoxic preconditioning also changes collagen deposition and inflammation in nanofat. Accordingly, we measured the numbers of rolling and adherent leukocytes in venules next to nanofat-seeded dermal substitutes using intravital fluorescence microscopy. Furthermore, the immune cell infiltration as well as Col I and Col III content of the implanted dermal substitutes were assessed by means of histology and immunohistochemistry. However, no marked differences could be detected in dermal substitutes seeded with hypoxic preconditioned nanofat when compared to controls. Hence, it can be excluded that the reduced vascularization capacity of hypoxic preconditioned nanofat has been induced by an altered fibrosis or inflammatory activity. Rather, as previously assumed, it most likely resulted from the excessive cumulative exposure to hypoxia during the 24 h in vitro preconditioning period followed by the subsequent in vivo hypoxic environment in the initial avascular phase after implantation of the dermal substitutes.

## 5. Conclusions

This study demonstrates that hypoxic preconditioning at 1% O_2_ for 24 h cannot be recommended for enhancing the regenerative in vivo vascularization capacity of nanofat. However, it should be considered that we only tested a single preconditioning protocol with a very low O_2_ concentration and rather long duration of hypoxia in combination with a specific dorsal skinfold chamber implantation model. Hence, our approach may have been too stressful for the nanofat, carrying the risk of transitioning potential beneficial effects of preconditioning into cell damage and death. Therefore, our findings should not be generalized. Instead, milder preconditioning protocols with shorter periods of hypoxia (e.g., 6 or 12 h) or higher oxygen levels (e.g., 5% O_2_) should be alternatively tested in future studies to achieve more favorable results. In doing so, it may be highly interesting to perform proteomic and lipidomic profiling of nanofat, as previously described [45,46], to analyze the effects of hypoxic preconditioning on its regenerative capacity at a molecular level. Moreover, because nanofat is a heterogeneous mixture of many different cell types, including stem cells, vascular cells and immune cells, sophisticated single-cell multi-omics analyses may give additional insights into cell-specific responses to hypoxia and oxidative stress [47,48].

## Figures and Tables

**Figure 1 cells-15-00100-f001:**
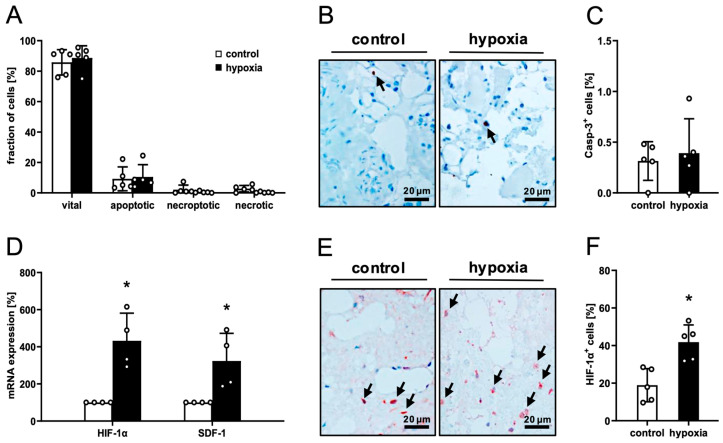
Ex vivo analysis of nanofat. (**A**) Fraction (%) of vital, apoptotic, necroptotic and necrotic cells in non-preconditioned (control; white bars, *n* = 4) and preconditioned nanofat (hypoxia; black bars, *n* = 4), as assessed by flow cytometry. Mean ± SEM. (**B**,**C**) Immunohistochemical detection of Casp-3^+^ apoptotic cells ((**B**), arrows) and their quantification (**C**) in non-preconditioned (control; white bar, *n* = 4) and preconditioned nanofat (hypoxia; black bar, *n* = 4). Means ± SEM. (**D**) HIF-1α and SDF-1 expression in non-preconditioned (control; white bars, *n* = 4) and preconditioned nanofat (hypoxia; black bars, *n* = 4). Means ± SEM; * *p* < 0.05 vs. control. (**E**,**F**) Immunohistochemical detection of HIF-1α^+^ cells ((**E**), arrows) and their quantification (**F**) in non-preconditioned (control; white bar, *n* = 4) and preconditioned nanofat (hypoxia; black bar, *n* = 4). Means ± SEM; * *p* < 0.05 vs. control.

**Figure 2 cells-15-00100-f002:**
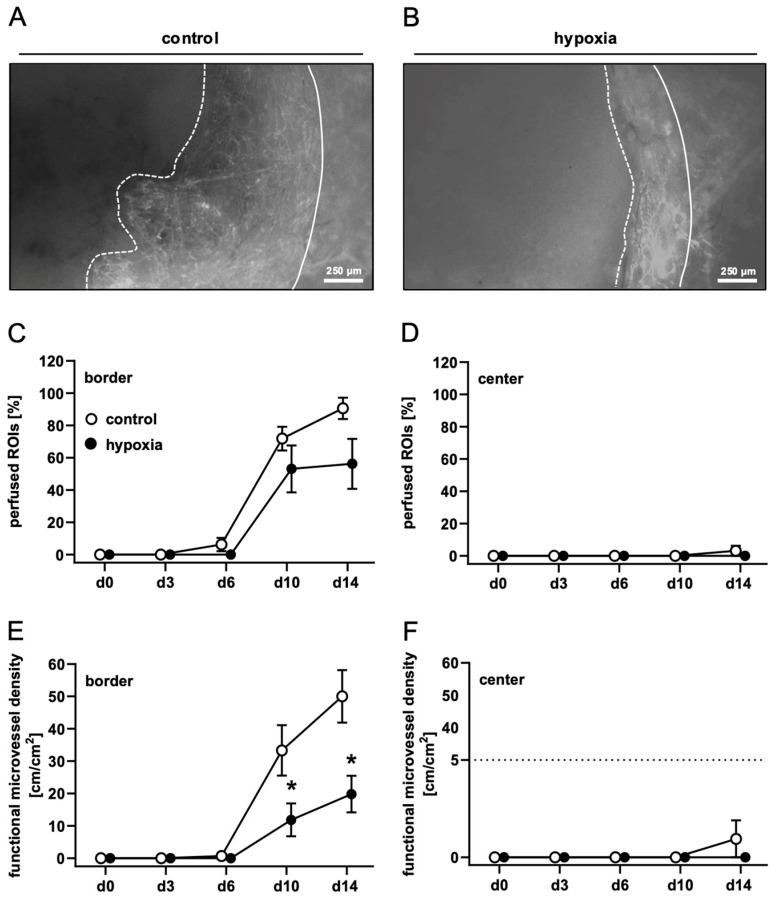
In vivo microscopy of nanofat-seeded dermal substitutes. (**A**,**B**) Intravital fluorescence microscopy of dermal substitutes seeded with non-preconditioned ((control, (**A**) and preconditioned nanofat ((hypoxia, (**B**) on day 14 (implant border = closed line; border of non-vascularized implant area = broken line). (**C**–**F**) Perfused ROIs (%) (**C**,**D**) and functional microvessel density (cm/cm^2^) (**E**,**F**) in the border (**C**,**E**) and center zones (**D**,**F**) of dermal substitutes seeded with non-preconditioned (control; white circles, *n* = 8) and preconditioned nanofat (hypoxia; black circles, *n* = 8) throughout the 14-day observation period, as assessed by intravital fluorescence microscopy. Means ± SEM. * *p* < 0.05 vs. control.

**Figure 3 cells-15-00100-f003:**
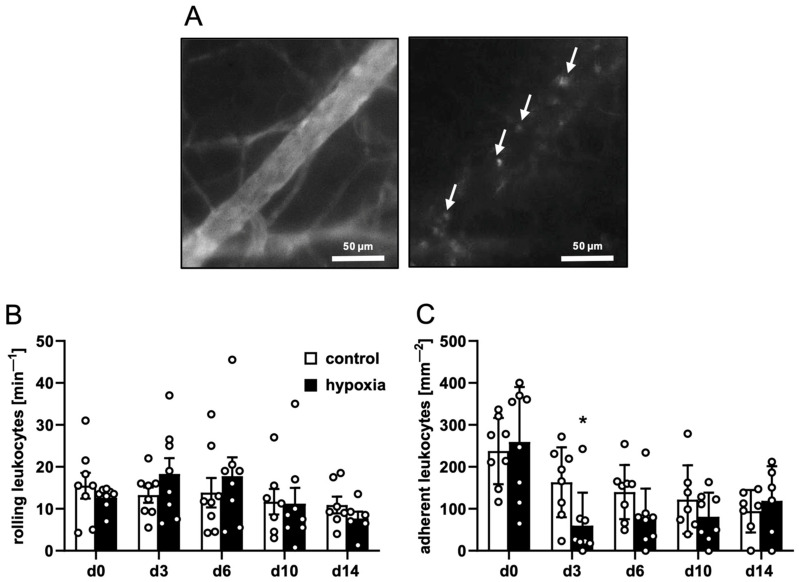
Leukocyte–endothelial cell interactions in response to nanofat-seeded dermal substitutes. (**A**) Intravital fluorescence microscopy of a collecting venule in direct vicinity to a dermal substitute seeded with non-preconditioned nanofat on day 3 (blue light epi-illumination, contrast enhancement by 5% FITC-labeled dextran (left panel); green light epi-illumination, in situ staining of leukocytes with 0.1% rhodamine 6G (right panel); white arrows = leukocytes). (**B**,**C**) Rolling leukocytes (min^−1^) (**B**) and adherent leukocytes (mm^−2^) (**C**) within postcapillary and collecting venules in direct vicinity to dermal substitutes seeded with non-preconditioned (control; white bars, *n* = 8) and preconditioned nanofat (hypoxia; black bars, *n* = 8) throughout the 14-day observation period, as assessed by intravital fluorescence microscopy. Means ± SEM. * *p* < 0.05 vs. control.

**Figure 4 cells-15-00100-f004:**
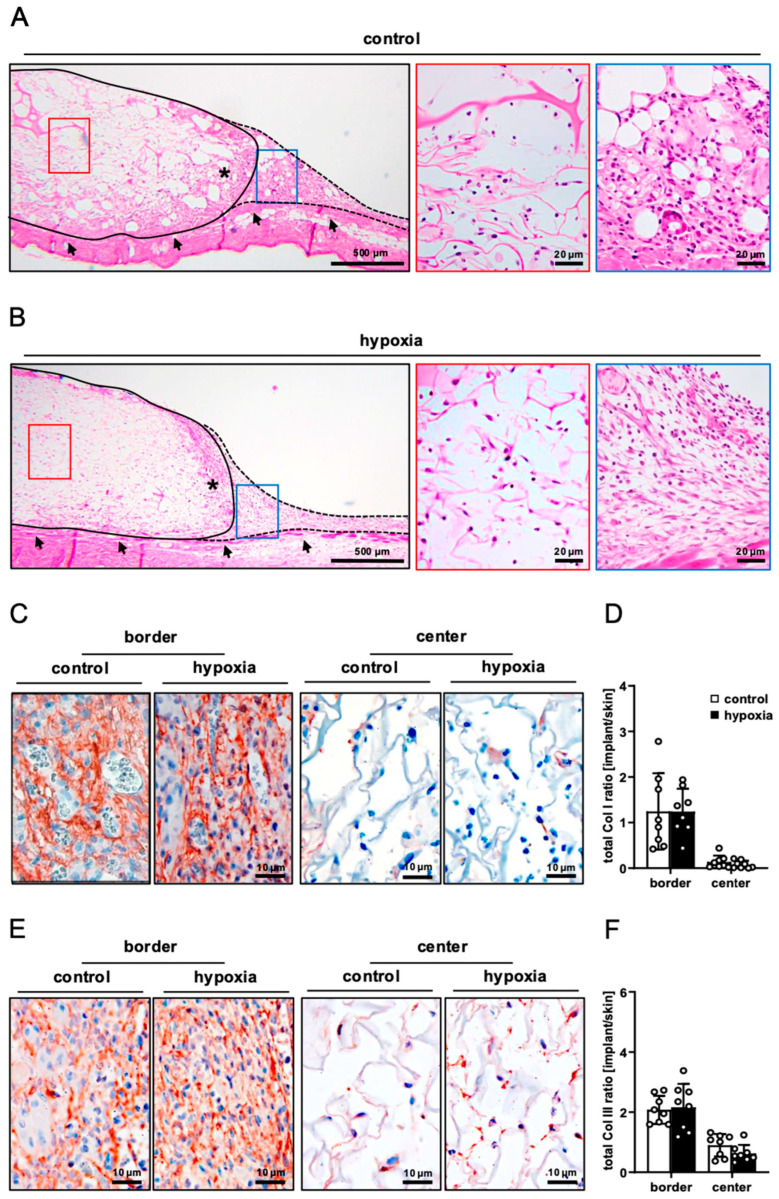
Tissue integration of nanofat-seeded dermal substitutes. (**A**,**B**) HE-stained sections of dermal substitutes seeded with non-preconditioned ((control, (**A**)) and preconditioned nanofat ((hypoxia, (**B**)) on day 14 after implantation (implant border = closed line; border zone = broken line; ROIs in the border and center zones of the implants shown in higher magnification = blue and red frame; panniculus carnosus muscle = arrows; granulation tissue = asterisks). (**C**,**E**) Immunohistochemical detection of Col I (**C**) and III (**E**) in the border and center zones of dermal substitutes seeded with non-preconditioned (control) and preconditioned nanofat (hypoxia) on day 14. (**D**,**F**) Total Col I (**D**) and Col III (**F**) ratio (implant/skin) in the border and center zones of dermal substitutes seeded with non-preconditioned (control; white bars, *n* = 8) and preconditioned nanofat (hypoxia; black bars, *n* = 8) on day 14, as assessed by immunohistochemistry. Means ± SEM.

**Figure 5 cells-15-00100-f005:**
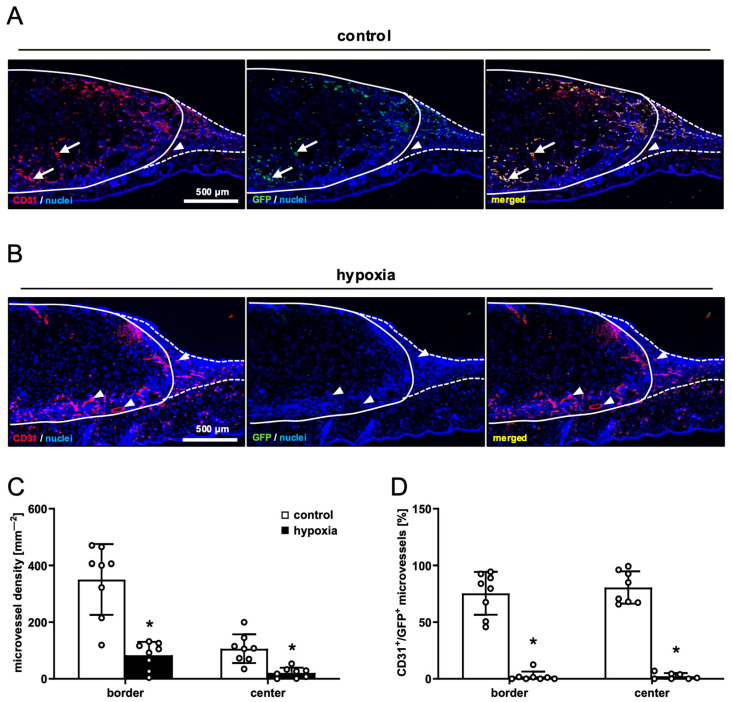
Vascularization of nanofat-seeded dermal substitutes. (**A**,**B**) Immunohistochemical detection of CD31^+^/GFP^−^ (arrowheads) and CD31^+^/GFP^+^ (arrows) microvessels in dermal substitutes seeded with non-preconditioned ((control, (**A**)) and preconditioned nanofat ((hypoxia, (**B**)) on day 14 (implant border = closed line; border zone = broken line). (**C**) Microvessel density (mm^−2^) of dermal substitutes seeded with non-preconditioned (control; white bars, *n* = 8) and preconditioned nanofat (hypoxia; black bars, *n* = 8) on day 14, as assessed by immunohistochemistry. Means ± SEM. * *p* < 0.05 vs. control. (**D**) CD31^+^/GFP^+^ microvessels (%) in the border and center zones of dermal substitutes seeded with non-preconditioned (control; white bars, *n* = 8) and preconditioned nanofat (hypoxia; black bars, *n* = 8) on day 14, as assessed by immunohistochemistry. Means ± SEM. * *p* < 0.05 vs. control.

**Figure 6 cells-15-00100-f006:**
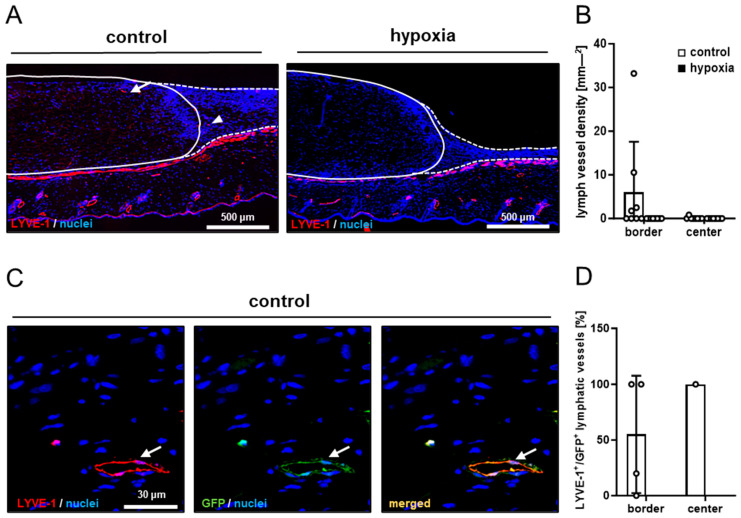
Lymph vessels in nanofat-seeded dermal substitutes. (**A**) Immunohistochemical detection of LYVE-1^+^ lymph vessels in the border zones (arrowhead) and the center (arrow) of dermal substitutes seeded with non-preconditioned (control) and preconditioned nanofat (hypoxia) on day 14 (implant border = closed line; border zone = broken line). (**B**) Lymph vessel density (mm^−2^) of dermal substitutes seeded with non-preconditioned (control; white bars, *n* = 8) and preconditioned nanofat (hypoxia; black bars, *n* = 8) on day 14, as assessed by immunohistochemistry. Means ± SEM. (**C**) Immunohistochemical detection of a LYVE-1^+^/GFP^+^ lymph vessel (arrow) in a dermal substitute seeded with non-preconditioned nanofat (control) on day 14. (**D**) LYVE-1^+^/GFP^+^ microvessels (%) in the border and center zones of dermal substitutes seeded with non-preconditioned (control; white bars, *n* = 1–4) and preconditioned nanofat (hypoxia; black bars, *n* = 0) on day 14, as assessed by immunohistochemistry. Means ± SEM.

**Figure 7 cells-15-00100-f007:**
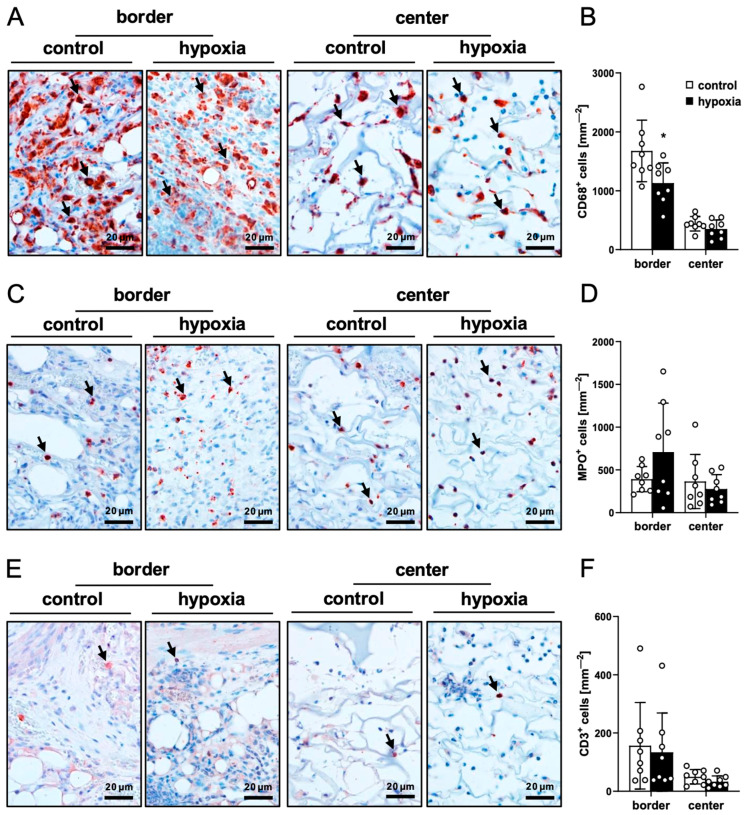
Ex vivo immune cell infiltration into nanofat-seeded dermal substitutes. (**A**,**C**,**E**) Immunohistochemical detection of CD68^+^ macrophages ((**A**), arrows), MPO^+^ neutrophilic granulocytes ((**C**), arrows) and CD3^+^ lymphocytes ((**E**), arrows) in the border and center zones of dermal substitutes seeded with non-preconditioned (control) and preconditioned nanofat (hypoxia) on day 14. (**B**,**D**,**F**) CD68^+^ macrophages (mm^−2^) (**B**), MPO^+^ neutrophilic granulocytes (mm^−2^) (**D**) and CD3^+^ lymphocytes (mm^−2^) (**F**) in the border and center zones of dermal substitutes seeded with non-preconditioned (control; white bars, *n* = 8) and preconditioned nanofat (hypoxia; black bars, *n* = 8) on day 14, as assessed by immunohistochemistry. Means ± SEM. * *p* < 0.05 vs. control.

**Table 1 cells-15-00100-t001:** Expression of pro- and anti-angiogenic proteins (% of control) in preconditioned nanofat, as assessed by a proteome profiler mouse angiogenesis array. Data are presented in a descending order as mean of two technical replicates.

Protein	Expression (% of Control)
** *Pro-angiogenic* **	
HGF	1027
Coagulator Factor III/Tissue Factor	436
MMP-8	285
GM-CSF	226
FGF basic/FGF-22	206
HB-EGF	195
Osteopontin	193
MMP-9	188
MIP-1alpha	182
IL-1alpha	167
Amphiregulin	161
IGFBP-1	160
Angiopoietin-1	153
Fractalkine/CX3CL1	148
PIGF-2	141
DLL4	141
IGFBP-3	133
KC/CXCL1/CINC-1/GRO-alpha	128
Angiogenin	127
VEGF/VPF	127
KGF/FGF-7	126
MMP-3	118
Cyr61/CCN1, IGFBP10	118
MCP-1/CCL2/JE	115
IL-10/CSIF	107
Leptin/OB	106
FGF acid/FGF-1/ECGF/HBGF-1	103
EGF	101
IL-1beta	99
SDF-1/CXCL12	96
CXCL 16	90
PD-ECGF	80
Proliferin	73
IGFBP-2	71
Endothelin-1	70
Endoglin/CD105	63
VEGF B/VRF	54
NOV/CCN3/IGFBP-9	48
PDGF-AA	35
** *Anti-angiogenic* **	
TIMP-1	231
IP-10/CXCL 10/CRG-2	128
Endostatin/Collagen VIII	118
TIMP-4	86
DPP IV/CD26	85
Serpin F1/PEDF	80
Serpin E1/PAI-1	72
Pentraxin-3/TSG-14	72
ADAMTS1/METH1	64
PDFG-AB/BB	63
Thrombospondin-2	59
Prolactin	51
Platelet facto 4/CXCL4	49
Angiopoietin-3	43

ADAMTS: A Disintegrin And Metalloproteinase with Thrombospondin Motifs; CCL: Chemokine (C-C motif) Ligand; CCN: Cellular Communication network factor; CD: Cluster of Differentiation; CINC: Cytokine-Induced Neutrophil Chemoattractant; CRG: Cytokine-Responsive Gene; CSIF: Cytokine Synthesis Inhibitory Factor; CX3CL: Chemokine (C-X3-C motif) Ligand; CXCL: Chemokine (C-X-C motif) Ligand; Cyr: Cysteine-Rich Angiogenic Inducer; DLL: Delta-Like Ligand; DPP: Dipeptidyl Peptidase; ECGF: Endothelial Cell Growth Factor; EGF: Epidermal Growth Factor; FGF: Fibroblast Growth Factor; GM-CSF: Granulocyte–Macrophage Colony-Stimulating Factor; GRO: Growth-Related Oncogene; HB-EGF: Heparin-Binding Epidermal Growth Factor; HBGF: Heparin-Binding Growth Factor; HGF: Hepatocyte Growth Factor; IGFBP: Insulin-Like Growth Factor Binding Protein; IL: Interleukin; IP: Interferon Gamma-Inducible Protein; KC: Keratinocyte Chemoattractant; KGF: Keratinocyte Growth Factor; MCP: Monocyte Chemoattractant Protein; MIP: Major Intrinsic Protein; MMP: Matrix Metalloproteinase; NOV: Nephroblastoma Overexpressed; OB: Obese; PAI: Plasminogen Activator Inhibitor; PD-ECGF: Platelet-Derived Endothelial Cell Growth Factor; PDGF: Platelet-Derived Growth Factor; PEDF: Pigment Epithelium-Derived Factor; PIGF: Placental Growth Factor; SDF: Stromal Cell-Derived Factor; TIMP: Tissue Inhibitor of Metalloproteinases; TSG: Tumor Necrosis Factor-Induced Protein; VEGF: Vascular Endothelial Growth Factor; VPF: Vascular Permeability Factor; VRF: Vascular Remodeling Factor.

**Table 2 cells-15-00100-t002:** Diameter (µm), centerline RBC velocity (µm/s), shear rate (s^−1^) and volumetric blood flow (pL/s) of microvessels within the border and center zones of dermal substitutes seeded with non-preconditioned (control; *n* = 8) and preconditioned nanofat (hypoxia; *n* = 8). Mean ± SEM. No significant differences.

	d0	d3	d6	d10	d14
***diameter* (µm):**					
border: control	-	-	22.5 ± 6.5	19.6 ± 1.1	15.2 ± 0.9
hypoxia	-	-	-	18.6 ± 1.0	15.5 ± 1.3
center: control	-	-	-	-	-
hypoxia	-	-	-	-	-
***centerline RBC velocity* (µm/s):**					
border: control	-	-	60.0 ± 30.0	93.5 ± 14.8	151.2 ± 18.4
hypoxia	-	-	-	84.1 ± 8.9	109.9 ± 16.9
center: control	-	-	-	-	-
hypoxia	-	-	-	-	-
***shear rate* (s^−1^):**					
border: control	-	-	19.9 ± 4.9	41.8 ± 6.6	97.7 ± 18.8
hypoxia	-	-		36.3 ± 3.5	65.9 ± 15.0
center: control	-	-	-	-	-
hypoxia	-	-	-	-	-
***volumetric blood flow* (pL/s):**					
border: control	-	-	20.5 ± 16.7	20.2 ± 4.1	18.0 ± 3.4
hypoxia	-	-		15.8 ± 2.3	12.1 ± 0.8
center: control	-	-	-	-	-
hypoxia	-	-	-	-	-

**Table 3 cells-15-00100-t003:** Diameter (µm), centerline RBC velocity (µm/s), shear rate (s^−1^) and volumetric blood flow (pL/s) of postcapillary and collecting venules in the direct vicinity of dermal substitutes seeded with non-preconditioned (control; *n* = 8) and preconditioned nanofat (hypoxia; *n* = 8). Mean ± SEM. * *p* < 0.05 vs. control.

	d0	d3	d6	d10	d14
***diameter* (µm):**					
control	41.5 ± 2.2 *	36.6 ± 0.3	38.2 ± 1.4	35.1 ± 0.9	34.5 ± 1.1
hypoxia	35.8 ± 0.6 *	34.8 ± 1.1	33.7 ± 0.9	33.7 ± 0.9	33.2 ± 1.0
***centerline RBC velocity* (µm/s):**					
control	516.5 ± 80.6	490.1 ± 55.9	646.9 ± 80.0	526.8 ± 86.3	562.8 ± 111.7
hypoxia	726.5 ± 101.7	471.5 ± 66.8	516.4 ± 81.8	445.5 ± 96.3	370.5 ± 53.4
***shear rate* (s^−1^):**					
control	100.7 ± 15.3	106.7 ± 12.2	139.7 ± 17.6	120.0 ± 21.1	128.3 ± 27.3
hypoxia	166.6 ± 24.9 *	108.2 ± 16.3	122.4 ± 18.6	102.9 ± 21.7	88.3 ± 12.4
***volumetric blood flow* (pL/s):**					
control	478.4 ± 110.0	462.2 ± 37.5	516.7 ± 93.4	336.0 ± 56.4	372.7 ± 64.3
hypoxia	448.5 ± 58.8	298.4 ± 44.4	319.9 ± 58.6	274.6 ± 63.6	239.0 ± 40.7

## Data Availability

Data is contained within the article.

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
