# Peer review of "Hypoxic Preconditioned Nanofat at 1% O2 for 24 h Loses Its Regenerative In Vivo Vascularization Capacity"

_cells, 2026, doi:10.3390/cells15020100_

Round 1

Reviewer 1 Report

Comments and Suggestions for Authors

This manuscript explores the effects of hypoxic preconditioning (1% O2, 24 h) on the viability, gene expression, and in vivo regenerative vascularization capacity of mouse-derived nanofat. Through ex vivo experiments, the study observed that hypoxic treatment significantly upregulates the expression of HIF-1a and SDF-1 and induces a pro-angiogenic protein expression profile; however, in the in vivo experiments using a dorsal skinfold chamber model, the authors unexpectedly found that the functional microvessel density in the hypoxic preconditioning group was significantly lower than that of the control group. Based on this, the authors suggest that the current hypoxic preconditioning protocol is not suitable for enhancing the in vivo regenerative vascularization capacity of nanofat. Nevertheless, there are still some deficiencies in the manuscript.

Major Comments

  1. In the discussion section, the authors should explore more deeply the potential biological mechanisms underlying the contradiction between the ex vivo pro-angiogenic phenotype and the decreased in vivo vascularization capacity. Although the experimental results showed that hypoxic treatment upregulated factors such as HIF-1a, the nanofat may have accumulated metabolic stress or pro-inflammatory factors during the preconditioning process, thereby inhibiting early vascular ingrowth after in vivo implantation. The authors should consider supplemental experiments or citing literature to analyze this sustained effect after preconditioning.
  2. Regarding the setting of experimental groups, the authors could consider adding comparison groups with different hypoxia concentrations (e.g., 5% O2) or different time gradients (e.g., 6 h, 12 h). Currently, using only 1% O2 for 24 h may represent a relatively strong stress, carrying the risk of preconditioning transitioning into cell damage. The authors should explain the rationale for selecting this specific parameter in the text or confirm whether a milder and more effective preconditioning window exists through supplemental experiments.
  3. Regarding the choice of statistical methods, the authors should clearly state whether corrections for multiple comparisons were performed when using Student's t-test or the Mann-Whitney test. Considering the analysis of multiple time points and multiple indicators, using one-way or two-way analysis of variance (ANOVA) might be more rigorous.
  4. Furthermore, the use of citations in the introduction and discussion sections can be further optimized. Given the breadth of research on nanofat and microcirculation, it is essential to integrate current multi-omics analysis and interdisciplinary technologies to understand microenvironmental changes in the context of tissue regeneration, which would help in more comprehensively evaluating the impact of hypoxia on adipose derivatives (doi: 10.1002/mdr2.70007, 10.34133/research.0387, 10.1016/j.cpan.2025.02.004).

Minor Comments

  1. The article requires professional native-speaker polishing to reduce grammatical errors and some unidiomatic expressions.

Author Response

Review of the manuscript ID cells-4077959 by Bonomi et al.

Reply to the comments of reviewer 1

We appreciate the fair and constructive comments of the reviewer. In the following, please find our point-by-point reply.

Major comments:

  1. Reviewer comment: In the discussion section, the authors should explore more deeply the potential biological mechanisms underlying the contradiction between the ex vivo pro-angiogenic phenotype and the decreased in vivo vascularization capacity. Although the experimental results showed that hypoxic treatment upregulated factors such as HIF-1a, the nanofat may have accumulated metabolic stress or pro-inflammatory factors during the preconditioning process, thereby inhibiting early vascular ingrowth after in vivo implantation. The authors should consider supplemental experiments or citing literature to analyze this sustained effect after preconditioning.

Reply: According to the comment of the reviewer, we have included a novel paragraph in the revised manuscript version, which discusses in more detail harmful effects of prolonged hypoxia as potential factors for the reduced in vivo vascularization capacity of the nanofat-seeded implants. This paragraph reads as follows:

‘In line with this view, Jian et al. [37] reported that exposure of endothelial progenitor cells to hypoxia for 24 h promotes their motility and tube formation, while these beneficial effects are reversed by prolonged hypoxia for 48 and 72 h. Moreover, it is well known that long hypoxic phases are typically associated with mitochondrial dysfunction, increased reactive oxygen species (ROS) formation and the accumulation of deleterious metabolites, resulting in cellular injury and final cell death [38]. An excessive activity of the HIF pathway may have additionally led to maladaptive outcomes, such as the induction of apoptosis [39]. In fact, the effects of HIF‑1 signaling have been described as a double‑edged sword, which may be particularly dependent on the degree and duration of hypoxia [39].’

(See lines 427-436 and 596-602; marked in yellow)   

New references:

  1. Jian, K.T.; Shi, Y.; Zhang, Y.; Mao, Y.M.; Liu, J.S.; Xue, F.L. Time course effect of hypoxia on bone marrow-derived endothelial progenitor cells and their effects on left ventricular function after transplanted into acute myocardial ischemia rat. Eur Rev Med Pharmacol Sci 2015, 19, 1043-1054.
  2. Vázquez-Galán, Y.I.; Guzmán-Silahua, S.; Trujillo-Rangel, W.Á.; Rodríguez-Lara, S.Q. Role of Ischemia/Reperfusion and Oxidative Stress in Shock State. Cells 2025, 14, DOI: 10.3390/cells14110808
  3. Zhang, Z.; Yao, L.; Yang, J.; Wang, Z.; Du, G. PI3K/Akt and HIF-1 signaling pathway in hypoxia-ischemia (Review). Mol Med Rep 2018, 18, 3547-3554. DOI: 10.3892/mmr.2018.9375

In addition, we have added an additional concluding sentence to the following paragraph to further support our assumption by our own findings on GFP+ microvessels and lymphatic vessels inside the nanofat (see lines 448-450; marked in yellow). 

  1. Reviewer comment: Regarding the setting of experimental groups, the authors could consider adding comparison groups with different hypoxia concentrations (e.g., 5% O2) or different time gradients (e.g., 6 h, 12 h). Currently, using only 1% O2 for 24 h may represent a relatively strong stress, carrying the risk of preconditioning transitioning into cell damage. The authors should explain the rationale for selecting this specific parameter in the text or confirm whether a milder and more effective preconditioning window exists through supplemental experiments.

Reply: The rationale for selecting the hypoxic preconditioning protocol with 1% O2 for 24 h is explained in the discussion section of our manuscript, which reads as follows:

‘Commonly tested hypoxic preconditioning protocols involve either intermittent or continuous exposure to low oxygen levels, with specific parameters varying depending on the target tissue and intended application. The most frequently applied protocols involve oxygen concentrations between 1% and 5% for various durations ranging from 10 min to 48 h [29-33]. Notably, a 24-h exposure to 1% oxygen has often been used for the preconditioning of stem cells [30, 34-36]. Accordingly, we also used this approach for the preconditioning of nanofat, which is known to be a rich source of ASCs.’

(See lines 405-411; marked in yellow)

Moreover, we have added a final conclusions section to our revised manuscript, which clearly addresses the important comment of the reviewer. This section reads as follows:

‘This study demonstrates that hypoxic preconditioning at 1% O2 for 24 h cannot be recommended for enhancing the regenerative in vivo vascularization capacity of nanofat. However, it should be considered that we herein only tested a single preconditioning protocol with a very low O2 concentration and rather long duration of hypoxia in combination with a specific dorsal skinfold chamber implantation model. Hence, our approach may have been too stressful for the nanofat, carrying the risk of transitioning potential beneficial effects of preconditioning into cell damage and death. Therefore, our findings should not be generalized. Instead, milder preconditioning protocols with shorter periods of hypoxia (e.g. 6 or 12 h) or higher oxygen levels (e.g. 5% O2) should be alternatively tested in future studies to achieve more favorable results.’

(See lines 471-481; marked in yellow)

In addition, we have added the following sentence at the end of the revised abstract: “Therefore, milder preconditioning protocols with shorter periods of hypoxia or higher oxygen levels should be alternatively tested in future studies.” (See lines 43-45, marked in yellow).

Finally, we have changed the general title “Hypoxic preconditioned nanofat loses its regenerative in vivo vascularization capacity” into the more specific title “Hypoxic preconditioned nanofat at 1% O2 for 24 h loses its regenerative in vivo vascularization capacity” (see lines 2-3; marked in yellow).

  1. Reviewer comment: Regarding the choice of statistical methods, the authors should clearly state whether corrections for multiple comparisons were performed when using Student's t-test or the Mann-Whitney test. Considering the analysis of multiple time points and multiple indicators, using one-way or two-way analysis of variance (ANOVA) might be more rigorous.

Reply: All our histological and immunohistochemical analyses were only performed at one time point (i.e. on day 14 after implantation of nanofat-seeded dermal substitutes). In contrast, the intravital fluorescence microscopic analyses were repeatedly performed over time. However, because we were not primarily interested in time effects but mainly differences between the two groups, we did not perform a statistical analysis of multiple time points and indicators (ANOVA). Accordingly, we did not perform corrections for multiple comparisons, which is now clearly stated in the statistics section of the revised version of our manuscript, as suggested by the reviewer (see line 214; marked in yellow). This type of analysis has also been performed in previously published manuscripts using the identical experimental approach [1,2].

References:

  1. Bonomi, F.; Limido, E.; Weinzierl, A.; Ampofo, E.; Harder, Y.; Menger, M.D.; Laschke, M.W. Nanofat Improves Vascularization and Tissue Integration of Dermal Substitutes without Affecting Their Biocompatibility. J Funct Biomater 2024, 15, 294. DOI: 3390/jfb15100294
  2. Bonomi, F.; Limido, E.; Weinzierl, A.; Bickelmann, C.; Ampofo, E.; Harder, Y.; Menger, M.D.; Laschke, M.W. Heat Preconditioning of Nanofat Does Not Improve Its Vascularization Properties. Cells 2025, 14, 581. DOI: 3390/cells14080581

  1. Reviewer comment: Furthermore, the use of citations in the introduction and discussion sections can be further optimized. Given the breadth of research on nanofat and microcirculation, it is essential to integrate current multi-omics analysis and interdisciplinary technologies to understand microenvironmental changes in the context of tissue regeneration, which would help in more comprehensively evaluating the impact of hypoxia on adipose derivatives (doi: 10.1002/mdr2.70007, 10.34133/research.0387, 10.1016/j.cpan.2025.02.004).

Reply: According to the comment of the reviewer, we have added the following paragraph with new references on multi-omics analyses to the revised version of our manuscript:

‘In doing so, it may be highly interesting to perform proteomic and lipidomic profiling of nanofat, as previously described [45,46], to analyze the effects of hypoxic preconditioning on its regenerative capacity at a molecular level. Moreover, because nanofat is a heterogeneous mixture of many different cell types, including stem cells, vascular cells and immune cells, sophisticated single-cell multi-omics analyses may give additional insights into cell-specific responses to hypoxia and oxidative stress [47,48].’

(See lines 481-487 and 617-628; marked in yellow)

New references:

  1. Sanchez-Macedo, N.; McLuckie, M.; Grünherz, L.; Lindenblatt, N. Protein Profiling of Mechanically Processed Lipoaspirates: Discovering Wound Healing and Antifibrotic Biomarkers in Nanofat. Plast Reconstr Surg 2022, 150, 341e-354e. DOI: 10.1097/PRS.0000000000009345
  2. Grünherz, L.; Kollarik, S.; Sanchez-Macedo, N.; McLuckie, M.; Lindenblatt, N. Lipidomic Analysis of Microfat and Nanofat Reveals Different Lipid Mediator Compositions. Plast Reconstr Surg 2024, 154, 895e-905e. DOI: 10.1097/PRS.0000000000011335
  3. Ye, J.; Gao, X.; Huang, X.; Huang, S.; Zeng, D.; Luo, W.; Zeng, C.; Lu, C.; Lu, L.; Huang, H.; Mo, K.; Huang, J.; Li, S.; Tang, M.; Wu, T.; Mai, R.; Luo, M.; Xie, M.; Wang, S.; Li, Y.; Lin, Y.; Liang, R. Integrating Single-Cell and Spatial Transcriptomics to Uncover and Elucidate GP73-Mediated Pro-Angiogenic Regulatory Networks in Hepatocellular Carcinoma. Research (Wash D C) 2024, 7, 0387. DOI: 10.34133/research.0387
  4. Barry, C.P.; Talbo, G.H.; Beauglehole, A.; Ovchinnikov, D.; Munro, T.; Mahler, S.; Baker, K.; Nielsen, L.K.; Mercer, T.R.; Marcellin, E. Resolving Single-Cell Gene Expression by Pseudotemporal Integration of Transcriptomic and Proteomic Datasets. Mol Cell Proteomics 2025, 25, 101475. DOI: 10.1016/j.mcpro.2025.101475

Minor comments:

  1. Reviewer comment: The article requires professional native-speaker polishing to reduce grammatical errors and some unidiomatic expressions.

Reply: According to the comment of the reviewer, the manuscript has been carefully checked for grammatical errors and unidiomatic expressions.     

Reviewer 2 Report

Comments and Suggestions for Authors

In their manuscript, the authors examine the effects of hypoxic preconditioning on the vascularization capacity of nanofat. They conducted the study using the well-established dorsal skinfold chamber model. The study is technically sound and employs rigorous in vivo imaging approaches. The topic is relevant in regenerative medicine. However, the manuscript must address several concerns, including biological interpretation, before it can be considered for publication. The specific points are as follows.

Major points.

  1. Discrepancy between ex vivo pro-angiogenic profiles and impaired in vivo vascularization

The most striking finding of this study is the clear discrepancy between the strongly pro-angiogenic ex vivo profile of hypoxia-preconditioned nanofat and its markedly impaired vascularization capacity in vivo. While the authors appropriately acknowledge this contradiction and propose cumulative hypoxic exposure as a potential explanation, this interpretation remains speculative.

No direct experimental evidence is provided to demonstrate that hypoxic preconditioning results in irreversible functional impairment of nanofat-derived vascular-forming cells or microvascular fragments. As such, the central mechanistic explanation remains insufficiently substantiated.

The authors are encouraged to either provide additional experimental support for this hypothesis or to more clearly frame it as a plausible but untested explanation.

  1. Generalization of the conclusion may be too broad

The statement that hypoxic preconditioning at 1% O₂ for 24 h “cannot be recommended” may be overly general, as it is based on a single preconditioning protocol and one implantation model. The conclusions should more clearly reflect the protocol- and context-dependent nature of the findings.

Moreover, to render the conclusions of the present study adequately supported, direct comparison of multiple hypoxic preconditioning conditions—varying both oxygen concentration and exposure duration—is indispensable. Reliance on a single preconditioning protocol (1% O₂ for 24 h) does not allow differentiation between protocol-specific effects and general properties of hypoxic preconditioning. Therefore, generalization of the current findings beyond the specific conditions tested is not sufficiently justified at this stage.

Comments on the Quality of English Language

none.

Author Response

Review of the manuscript ID cells-4077959 by Bonomi et al.

Reply to the comments of reviewer 2

We appreciate the fair and constructive comments of the reviewer. In the following, please find our point-by-point reply.

Major points:

  1. Reviewer comment: Discrepancy between ex vivo pro-angiogenic profiles and impaired in vivo vascularization

The most striking finding of this study is the clear discrepancy between the strongly pro-angiogenic ex vivo profile of hypoxia-preconditioned nanofat and its markedly impaired vascularization capacity in vivo. While the authors appropriately acknowledge this contradiction and propose cumulative hypoxic exposure as a potential explanation, this interpretation remains speculative.

No direct experimental evidence is provided to demonstrate that hypoxic preconditioning results in irreversible functional impairment of nanofat-derived vascular-forming cells or microvascular fragments. As such, the central mechanistic explanation remains insufficiently substantiated.

The authors are encouraged to either provide additional experimental support for this hypothesis or to more clearly frame it as a plausible but untested explanation.

Reply: According to the comment of the reviewer, we now more clearly frame our statement of negative effects of hypoxic preconditioning in combination with prolonged in vivo hypoxia as an untested hypothesis (see line 424; marked in yellow).

Moreover, according to another comment of reviewer 1, we have added an additional paragraph in the revised manuscript version, which discusses in more detail harmful effects of prolonged hypoxia as potential factors for the reduced in vivo vascularization capacity of the analyzed nanofat-seeded implants to further support our hypothesis (see lines 427-436 and 596-602; marked in yellow). 

Finally, we feel that our hypothesis is not only speculative but indirectly also supported by our immunohistochemical results on GFP+ blood and lymphatic vessels within the nanofat-seeded implants. In fact, we found that dermal substitutes seeded with control nanofat contained ~80% GFP+ microvessels, indicating their origin from the seeded nanofat. On the other hand, implants seeded with preconditioned nanofat exhibited almost no GFP+ microvessels on day 14 after implantation into the dorsal skinfold chamber. In addition, we detected some LYVE-1+/GFP+ lymph vessels in control nanofat-seeded dermal substitutes, whereas the hypoxic preconditioned group did not contain any GFP+ lymph vessels. Based on these results, we have added the following concluding sentence at the end of this discussion paragraph of the revised manuscript version:

‘These observations further support our assumption that prolonged hypoxia may have destroyed the blood and lymphatic vessels inside the preconditioned nanofat through the mechanisms mentioned before.’

(See lines 448-450; marked in yellow)

  1. Reviewer comment: Generalization of the conclusion may be too broad

The statement that hypoxic preconditioning at 1% O for 24 h “cannot be recommended” may be overly general, as it is based on a single preconditioning protocol and one implantation model. The conclusions should more clearly reflect the protocol- and context-dependent nature of the findings.

Moreover, to render the conclusions of the present study adequately supported, direct comparison of multiple hypoxic preconditioning conditions—varying both oxygen concentration and exposure duration—is indispensable. Reliance on a single preconditioning protocol (1% O for 24 h) does not allow differentiation between protocol-specific effects and general properties of hypoxic preconditioning. Therefore, generalization of the current findings beyond the specific conditions tested is not sufficiently justified at this stage.

Reply: We agree with the reviewer that a generalization of our findings beyond the specifically applied preconditioning protocol with 1% O2 for 24 h is not justified. Therefore, we have performed the following changes in our revised manuscript version:

We have changed the general title “Hypoxic preconditioned nanofat loses its regenerative in vivo vascularization capacity” into the more specific title “Hypoxic preconditioned nanofat at 1% O2 for 24 h loses its regenerative in vivo vascularization capacity” (see lines 2-3; marked in yellow).

Moreover, we have added the following sentence at the end of the revised abstract: “Therefore, milder preconditioning protocols with shorter periods of hypoxia or higher oxygen levels should be alternatively tested in future studies.” (See lines 43-45, marked in yellow).

Finally, we have added a conclusion section at the end of our revised discussion, which clearly indicates that a generalization of our findings beyond the specifically applied preconditioning protocol with 1% O2 for 24 h is not justified. This section reads as follows:

‘This study demonstrates that hypoxic preconditioning at 1% O2 for 24 h cannot be recommended for enhancing the regenerative in vivo vascularization capacity of nanofat. However, it should be considered that we herein only tested a single preconditioning protocol with a very low O2 concentration and rather long duration of hypoxia in combination with a specific dorsal skinfold chamber implantation model. Hence, our approach may have been too stressful for the nanofat, carrying the risk of transitioning potential beneficial effects of preconditioning into cell damage and death. Therefore, our findings should not be generalized. Instead, milder preconditioning protocols with shorter periods of hypoxia (e.g. 6 or 12 h) or higher oxygen levels (e.g. 5% O2) should be alternatively tested in future studies to achieve more favorable results.’

(See lines 471-481; marked in yellow)

Reviewer 3 Report

Comments and Suggestions for Authors
  1. The manuscript uses a single hypoxic preconditioning protocol (1% O₂ for 24 h). Although this approach is supported by previous literature, the study would benefit from a more explicit discussion explaining why alternative conditions (e.g., 2–5% O₂ or shorter exposure times) were not explored, especially considering the negative in vivo outcome.
  2. The authors clearly demonstrate a pro-angiogenic activation ex vivo, while the in vivo results show an opposite effect. This discrepancy is very interesting, but the discussion could be strenghten by further elaboration on potential cellular mechanisms, such as functional impairment of microvascular fragments or metabolic alterations induced by prolonged hypoxia.

  3. The study shows a marked reduction of GFP⁺ microvessels derived from hypoxia-preconditioned nanofat. However, a direct functional assessment of microvascular fragment viability or integrity after hypoxic exposure is lacking and would support the central interpretation of the study.

  4. The proteome array reveals relevant changes in pro- and anti-angiogenic factors, but the data are mainly presented descriptively. Highlighting and discussing the most strongly altered factors with known relevance for functional angiogenesis in vivo would improve the impact of these results.

  5. While the sample size is consistent with previous studies using the dorsal skinfold chamber model, the manuscript would benefit from a brief justification of the sample size or a discussion regarding the statistical power to detect biologically meaningful differences.

  6. The conclusions clearly discourage the tested hypoxic protocol. However, expanding the final section to discuss how these findings may inform the development of safer or optimized preconditioning strategies for clinical application would increase the translational relevance.

Author Response

Review of the manuscript ID cells-4077959 by Bonomi et al.

Reply to the comments of reviewer 3

We appreciate the fair and constructive comments of the reviewer. In the following, please find our point-by-point reply.

  1. Reviewer comment: The manuscript uses a single hypoxic preconditioning protocol (1% O for 24 h). Although this approach is supported by previous literature, the study would benefit from a more explicit discussion explaining why alternative conditions (e.g., 2–5% O or shorter exposure times) were not explored, especially considering the negative in vivo outcome.

Reply: According to the comment of this reviewer and additional comments from the other two reviewers, we have performed the following changes in our revised manuscript version:

We have changed the general title “Hypoxic preconditioned nanofat loses its regenerative in vivo vascularization capacity” into the more specific title “Hypoxic preconditioned nanofat at 1% O2 for 24 h loses its regenerative in vivo vascularization capacity” (see lines 2-3; marked in yellow).

Moreover, we have added the following sentence at the end of the revised abstract: “Therefore, milder preconditioning protocols with shorter periods of hypoxia or higher oxygen levels should be alternatively tested in future studies.” (See lines 43-45, marked in yellow).

Finally, we have added a conclusion section at the end of our revised discussion, which clearly indicates that a generalization of our findings beyond the specifically applied preconditioning protocol with 1% O2 for 24 h is not justified. This section reads as follows:

‘This study demonstrates that hypoxic preconditioning at 1% O2 for 24 h cannot be recommended for enhancing the regenerative in vivo vascularization capacity of nanofat. However, it should be considered that we herein only tested a single preconditioning protocol with a very low O2 concentration and rather long duration of hypoxia in combination with a specific dorsal skinfold chamber implantation model. Hence, our approach may have been too stressful for the nanofat, carrying the risk of transitioning potential beneficial effects of preconditioning into cell damage and death. Therefore, our findings should not be generalized. Instead, milder preconditioning protocols with shorter periods of hypoxia (e.g. 6 or 12 h) or higher oxygen levels (e.g. 5% O2) should be alternatively tested in future studies to achieve more favorable results.’

(See lines 471-481; marked in yellow)

  1. Reviewer comment: The authors clearly demonstrate a pro-angiogenic activation ex vivo, while the in vivo results show an opposite effect. This discrepancy is very interesting, but the discussion could be strenghten by further elaboration on potential cellular mechanisms, such as functional impairment of microvascular fragments or metabolic alterations induced by prolonged hypoxia.

Reply: According to the comment of the reviewer, we have included a novel paragraph in the revised manuscript version, which discusses in more detail harmful effects of prolonged hypoxia as potential factors for the reduced in vivo vascularization capacity of the nanofat-seeded implants. This paragraph reads as follows:

‘In line with this view, Jian et al. [37] reported that exposure of endothelial progenitor cells to hypoxia for 24 h promotes their motility and tube formation, while these beneficial effects are reversed by prolonged hypoxia for 48 and 72 h. Moreover, it is well known that long hypoxic phases are typically associated with mitochondrial dysfunction, increased reactive oxygen species (ROS) formation and the accumulation of deleterious metabolites, resulting in cellular injury and final cell death [38]. An excessive activity of the HIF pathway may have additionally led to maladaptive outcomes, such as the induction of apoptosis [39]. In fact, the effects of HIF‑1 signaling have been described as a double‑edged sword, which may be particularly dependent on the degree and duration of hypoxia [39].’

(See lines 427-436 and 596-602; marked in yellow)   

New references:

  1. Jian, K.T.; Shi, Y.; Zhang, Y.; Mao, Y.M.; Liu, J.S.; Xue, F.L. Time course effect of hypoxia on bone marrow-derived endothelial progenitor cells and their effects on left ventricular function after transplanted into acute myocardial ischemia rat. Eur Rev Med Pharmacol Sci 2015, 19, 1043-1054.
  2. Vázquez-Galán, Y.I.; Guzmán-Silahua, S.; Trujillo-Rangel, W.Á.; Rodríguez-Lara, S.Q. Role of Ischemia/Reperfusion and Oxidative Stress in Shock State. Cells 2025, 14, DOI: 10.3390/cells14110808
  3. Zhang, Z.; Yao, L.; Yang, J.; Wang, Z.; Du, G. PI3K/Akt and HIF-1 signaling pathway in hypoxia-ischemia (Review). Mol Med Rep 2018, 18, 3547-3554. DOI: 10.3892/mmr.2018.9375

  1. Reviewer comment: The study shows a marked reduction of GFP microvessels derived from hypoxia-preconditioned nanofat. However, a direct functional assessment of microvascular fragment viability or integrity after hypoxic exposure is lacking and would support the central interpretation of the study.

Reply: We agree with the reviewer that it would have been interesting to selectively assess the viability of microvascular fragment viability or integrity after hypoxic exposure. Although such analyses were not performed in the present study, we feel that our results on GFP+ signals within the preconditioned nanofat support our main hypothesis that prolonged hypoxia may have destroyed the blood and lymphatic vessels inside the preconditioned nanofat. This is now clearly stated at the end of the corresponding paragraph of our revised manuscript version (see lines 448-450; marked in yellow). 

  1. Reviewer comment: The proteome array reveals relevant changes in pro- and anti-angiogenic factors, but the data are mainly presented descriptively. Highlighting and discussing the most strongly altered factors with known relevance for functional angiogenesis in vivo would improve the impact of these results.

Reply: The results of the proteome array should be interpreted as a simple screening approach to get a feeling about the angiogenic phenotype of the preconditioned nanofat. Accordingly, we agree with the reviewer that the data are mainly presented descriptively. However, we deliberately did not focus on individual altered factors, because this could lead to misinterpretations by the readers. In fact, we feel that the overall expression pattern of pro- and anti-angiogenic factors determines the final angiogenic activity of a tissue.

  1. Reviewer comment: While the sample size is consistent with previous studies using the dorsal skinfold chamber model, the manuscript would benefit from a brief justification of the sample size or a discussion regarding the statistical power to detect biologically meaningful differences.

Reply: According to the comment of the reviewer, we have added the following information to the statistical section of our revised manuscript:

‘The group sizes were chosen according to previous studies using the herein described model [27,28]. Following the 3R principle in animal testing, the number of animals per group was reduced to a minimum while guaranteeing sufficient statistical power (0.8) to detect biological meaningful differences.’

(See lines 211-214; marked in yellow)

References:

  1. Bonomi, F.; Limido, E.; Weinzierl, A.; Ampofo, E.; Harder, Y.; Menger, M.D.; Laschke, M.W. Nanofat Improves Vascularization and Tissue Integration of Dermal Substitutes without Affecting Their Biocompatibility. J Funct Biomater 2024, 15, 294. DOI: 3390/jfb15100294
  2. Bonomi, F.; Limido, E.; Weinzierl, A.; Bickelmann, C.; Ampofo, E.; Harder, Y.; Menger, M.D.; Laschke, M.W. Heat Preconditioning of Nanofat Does Not Improve Its Vascularization Properties. Cells 2025c, 14, 581. DOI: 3390/cells14080581

  1. Reviewer comment: The conclusions clearly discourage the tested hypoxic protocol. However, expanding the final section to discuss how these findings may inform the development of safer or optimized preconditioning strategies for clinical application would increase the translational relevance.

Reply: According to the comment of this reviewer and a comment of another reviewer, we have expanded the final conclusions section of our revised manuscript, which reads as follows:

‘This study demonstrates that hypoxic preconditioning at 1% O2 for 24 h cannot be recommended for enhancing the regenerative in vivo vascularization capacity of nanofat. However, it should be considered that we herein only tested a single preconditioning protocol with a very low O2 concentration and rather long duration of hypoxia in combination with a specific dorsal skinfold chamber implantation model. Hence, our approach may have been too stressful for the nanofat, carrying the risk of transitioning potential beneficial effects of preconditioning into cell damage and death. Therefore, our findings should not be generalized. Instead, milder preconditioning protocols with shorter periods of hypoxia (e.g. 6 or 12 h) or higher oxygen levels (e.g. 5% O2) should be alternatively tested in future studies to achieve more favorable results. In doing so, it may be highly interesting to perform proteomic and lipidomic profiling of nanofat, as previously described [45,46], to analyze the effects of hypoxic preconditioning on its regenerative capacity at a molecular level. Moreover, because nanofat is a heterogeneous mixture of many different cell types, including stem cells, vascular cells and immune cells, sophisticated single-cell multi-omics analyses may give additional insights into cell-specific responses to hypoxia and oxidative stress [47,48].’

(See lines 471-487 and 617-628; marked in yellow)

New references:

  1. Sanchez-Macedo, N.; McLuckie, M.; Grünherz, L.; Lindenblatt, N. Protein Profiling of Mechanically Processed Lipoaspirates: Discovering Wound Healing and Antifibrotic Biomarkers in Nanofat. Plast Reconstr Surg 2022, 150, 341e-354e. DOI: 10.1097/PRS.0000000000009345
  2. Grünherz, L.; Kollarik, S.; Sanchez-Macedo, N.; McLuckie, M.; Lindenblatt, N. Lipidomic Analysis of Microfat and Nanofat Reveals Different Lipid Mediator Compositions. Plast Reconstr Surg 2024, 154, 895e-905e. DOI: 10.1097/PRS.0000000000011335
  3. Ye, J.; Gao, X.; Huang, X.; Huang, S.; Zeng, D.; Luo, W.; Zeng, C.; Lu, C.; Lu, L.; Huang, H.; Mo, K.; Huang, J.; Li, S.; Tang, M.; Wu, T.; Mai, R.; Luo, M.; Xie, M.; Wang, S.; Li, Y.; Lin, Y.; Liang, R. Integrating Single-Cell and Spatial Transcriptomics to Uncover and Elucidate GP73-Mediated Pro-Angiogenic Regulatory Networks in Hepatocellular Carcinoma. Research (Wash D C) 2024, 7, 0387. DOI: 10.34133/research.0387
  4. Barry, C.P.; Talbo, G.H.; Beauglehole, A.; Ovchinnikov, D.; Munro, T.; Mahler, S.; Baker, K.; Nielsen, L.K.; Mercer, T.R.; Marcellin, E. Resolving Single-Cell Gene Expression by Pseudotemporal Integration of Transcriptomic and Proteomic Datasets. Mol Cell Proteomics 2025, 25, 101475. DOI: 10.1016/j.mcpro.2025.101475

Round 2

Reviewer 1 Report

Comments and Suggestions for Authors

The authors have satisfactorily addressed the previous queries. This version of manuscript is acceptable for the journal.

Reviewer 2 Report

Comments and Suggestions for Authors

I think the authors have addressed almost all of my concerns and I have no further points.

Comments on the Quality of English Language

none.